# Accessibility and Connectivity Criteria for Assessing Walkability: An Application in Qazvin, Iran

Mona Jabbari *, Fernando Fonseca and Rui Ramos 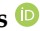

CTAC (Centre for Territory, Environment and Construction), University of Minho, Guimarães,
4710-057 Braga, Portugal; ffonseca@civil.uminho.pt (F.F.); rramos@civil.uminho.pt (R.R.)
* Correspondence: mona.jabbari@civil.uminho.pt

**Abstract:** Distance is a recognized key determinant of walking. Pedestrians tend to choose the shortest route between two points. Shortest routes can be spatially described in terms of distances between two points or topologically described as the number of turns/directional changes between these points. This paper presents a methodology to evaluate the conditions provided by a street network to pedestrians, by using two space syntax measures. Accessibility was calculated through Angular Segment Analysis by Metric Distance (ASAMeD), a measure of street integration and choice strongly correlated with pedestrian movement pattern. Street Connectivity was calculated by using the space syntax measure of connectivity, which shows the direct connection of street nodes to each individual nodes. The streets criterion values of both approaches were normalized by using fuzzy logic linear functions. The method was applied in the city center of Qazvin, Iran. Results showed that the urban structure of Qazvin has a strong impact on the performance of the network. The old neighborhood centers widespread in the city center presented a high topological accessibility, while the most connected street are those streets crossing and surrounding the neighborhood areas. The method can be used to evaluate and improve pedestrian networks, as it can distinguish the most and least attractive streets according to the criteria used. These findings can be used to guide policies towards improving walkability and to create more walkable and sustainable cities.

**Keywords:** pedestrian network; pedestrian movement pattern; space syntax; urban morphology

## 1. Introduction

Walking is one of the least expensive and most accessible modes of transportation. Nowadays, there is growing interest in walkability due to sustainable mobility being widespread in urban areas. Walking has become an important alternative means of transport for short trips in urban areas. Thus, an in-depth analysis is necessary to evaluate the quality of walking environments [1,2]. In recent years, several authors have analyzed walkability and focused on diverse topics such as (i) physical activity and health [3–6]; (ii) criteria affecting walking [7–12]; (iii) creating pedestrian indices or walkability scores [13–18]; and (iv) behavioral research [19,20].

Recently, behavioral researchers have focused on walking as an alternative means of transport and the criteria interfering with the choices people make when walking [1,4,21–23]. Despite comprehensive studies on pedestrian behavior, insufficient work has been carried out to examine the link between the built environment and pedestrian route choice [24–26]. Studies on the type of routes preferred by pedestrians can be used to improve the level of walkability in our cities [21,27]. Some studies have shown that pedestrians tend to choose routes based on the least directional change and the shortest distance to minimize walking time [2,20]. In addition, other researchers tried to understand in-depth the role of urban morphology in providing suitable pedestrian access and connect to urban spaces [24,28,29]. Nonetheless, it seems that no study in the space syntax literature has started to apply both (accessibility and connectivity) criteria simultaneously to assess their impacts on pedestrian movement patterns.

The space syntax method was developed as a tool for understanding spatial structure from the topological and geographical point of view [30,31]. In space syntax, the streets are represented in two ways: segment map and axial map [32].

The segment map splits the axial lines at junctions or intersections and considers the whole of the changing direction from an origin to a destination [33,34]. Accessibility refers to the ability to reach the place or destination and represents pedestrian movement patterns in the urban fabric [24,31,35–38].

An axial map represents lines of sight, e.g., represents the visibility including the longest and fewest axial lines in a street network. Physical distance does not necessarily act as a deterrent in this context; rather, it is the relative geographical association of streets through their common junctions or intersections, which provides a measure of distance [39]. Connectivity is one of the crucial aspects characterizing complex networks and is capable of making and controlling connections within two or more points in the spatial structure [40].

As pedestrians are more sensitive to their surrounding environment, numerous features and urban parameters influence their movement. Hence, numerous features of movement patterns and pedestrian flows in traditional cities have shaped the urban structure, namely, the street network configuration in the micro-scale [37,41]. Such an urban structure has strong effects on transportation, economic growth, social equity, sustainable urban development, and the quality of life of a city's inhabitants [42,43]. In the past, the urban structure became ever more polycentric, consisting of a complex hierarchy of different kinds of centers, neighborhoods, and communities that are connected by a multiplicity system [44–47].

Space syntax theory reflects human movement decision-making and urban morphology includes criteria such as street integration and choice, which have impacts on the urban planning and design process [48,49]. However, most previous research addressed built environment-physical activity studies [26,50]. Studies have also estimated pedestrian movement in the city. For example, Baran, Rodríguez [51] identified a positive association between total utilitarian walking and two of the syntactic measures (control and integration). Furthermore, studies have linked space syntax with pedestrian behavior through travel surveys [51], but without including urban morphology criteria in the analysis. The transformations raise the important question of how to engage accessibility and connectivity as the main criteria of urban morphology to the planning process for walkability. It seems that in the future, by responding to this question will improve urban planning processing and lead cities to a new structure for pedestrians walkability.

This study aims to understand the relationships between urban morphology and walkability by combining geographic and topological measures and by using qualitative and quantitative approaches. The method was applied in the city center of Qazvin, Iran. Qualitatively, a documentary method was applied to find some a normative framework for the urban morphology pattern in Qazvin's traditional fabric. In fact, the historical documentary method presents information to compare and contrast the exact urban conditions in the past and now [52]. Quantitatively the method was supported, in terms of accessibility and connectivity, by streets analyses within a Geographic Information System (GIS) and by space syntax to assess the condition that streets network provided to pedestrians. Two space syntax analysis were performed: (i) an Angular Segment Analysis by Metric Distance (ASAMeD) and street connectivity. Accessibility was calculated through ASAMeD, a measure strongly correlated with pedestrian movement pattern, which shows the opportunity and the extent to which streets can attract or deter pedestrian movement. Street connectivity was measured by using the space syntax measure of connectivity which shows the direct connection of street nodes to each individual node. The values of both geographic and topological approaches were normalized by using fuzzy logic linear functions so they can be compared and combined. Finally, spaces and streets are more and less conducive for walking according to the criteria adopted. So, the streets have been ranking according to their level of accessibility and connectivity and showed in digital maps.

The research considered the highest-ranking of the streets from two analyses (the results of ASAMeD and street connectivity) through GIS (Geographic Information System) in order to assess the condition of the pedestrian network in Qazvin city. Planners and designers need to develop a deep and rich understanding of Qazvin's inherent urban structure and how the urban structure influences the condition of walkability.

The rest Sections of this paper are structured as follows. After a review of literature in Sections 2–4 describe the method and the case study. Likewise, in Section 5, the outputs of the study are presented as results. Sections 6 and 7, put forward the discussion and conclusions.

## 2. Literature Studies

The urban structure has been shaped from the interaction between human behaviors and built environments [26,53,54]. More particularly, urban morphology as part of the urban structure reflects the city's layout, entailing the spatial analysis of street patterns, buildings, and open spaces [35,55]. Urban morphology provides information about the city's structural characteristics including the structural origins and impacts of historical change on the chronological processes concerning the construction and reconstruction of a city [56]. It is mostly classified according to the way streets and buildings are arranged. Thus, the city morphology is determined by various planning regulations and by the socioeconomic conditions verified over time. The study in [56] reports over 100 descriptors to characterize urban street patterns including radial-concentric, rectangular, grid, irregular, star, and satellite, among many other forms.

One of the tools the research applies is space syntax, which is used to frequently analyze the influence of spatial configuration of urban systems on a wide range of issues including walking [31,57]. The technique appeared in the 1970s [58] and was mostly fostered by the works of Hillier and Hanson [59] and Hillier, Penn [60]. Space syntax is a combination of theories, methods, and tools for urban design and offers the possibility of estimating the theoretical accessibility and connectivity in urban morphology [61].

### 2.1. Overview to Accessibility

Accessibility is based on the social/spatial relationship that relates one space to another place in an urban system [21,62–64]. In other words, one area consisting of more integrated streets, which are likely to be more accessible from other areas, will draw more pedestrians. A positive correlation between higher street integration and a greater pedestrian volume has been found in several previous studies [33,38,65–67]. Places linked directly to other environments are more accessible and tend to attract more people making areas busier [51,68]. The space syntax theory considers that accessibility depends on people's wayfinding skills and mental conceptualizations of the environment. The most accessible locations are not necessarily those closest to all other locations, but those closest in terms of topological turns Hillier and Vaughan [58]. The inclusion of the angular dimension through angular segment analysis (ASA) in the space syntax happened in the early 2000s by Dalton and Turner [62]. ASA breaks axial lines into segments, and then records the sum of the angles turned from the starting segment to any other segment within the network. In such an approach, distance is measured in terms of angular "costs". For the minimum turn ($0°$), the distance cost value is 0; for the maximum turn ($180°$), the cost value is 2. The distance cost values for the rest of the turn angles ($\alpha$) can be calculated using the formula: $V(\alpha) = \alpha/180 \times 2$ ($\alpha$ is expressed in the degree unit). The geometric distance is, thus, based on the smallest angular changes and has been called Angular Segment Analysis-ASA [69].

Some researchers have found that a geometric or least angle analysis within a metric catchment area is an excellent predictor of movement [33,41], called ASAMeD (angular segment analysis by metric distance), which was combined with closeness and betweenness indicators. Consequently, ASAMeD analysis represents aspects to the spatial pattern. The result of the ASAMeD analysis strongly correlates with pedestrian movement

pattern [59,70–72]. Hence, ASAMeD identifies potential streets that pedestrian use for walking. This is based on the routes' position in the urban fabric in terms of closeness and betweenness, which are more accessible in the urban structure [65,73].

### 2.2. Overview to Connectivity

Over the years, street connectivity has contributed to a greater understanding of the spatial configuration of street networks and the location of economic activities [60,72]. Street connectivity can be understood as the directness and availability of alternative routes between destinations and that can attract or deter pedestrian movement opportunity. Hence, a well-connected network increase walkability in two ways: more interconnected streets provide more potential routes and, at the same time, the pedestrians can reach easily more destinations. Street connectivity is often described by measurable properties of the street network. It has been described by a considerable number of different attributes, mostly by calculating ratios, such as intersection and street densities. Intersection density has been widely used as a descriptor of street connectivity. Graphically, it represents the number of road intersections in a specific area [30,74–76]. Connectivity analysis focuses on the spatial relationships between streets within a network [71]. Therefore, streets connectivity seems to be one of the important planning processes that affect different areas of a city in terms of economic, ecological, and culture [77–79].

### 3. Methodology

This paper presents a methodology to evaluate the conditions provided by a street network to pedestrians, by combining two space syntax and topological measures associated to urban morphology. The study extracted the main morphological features through a cognitive recognition experiment and then established a quantitative description method for them. The methodological procedures adopted in the study are as follows: (i) a historical documentary method to understand the urban morphology of Qazvin, the case study; (ii) space syntax analysis to evaluate street connectivity and accessibility criterion; and (iii) normalization of the street criterion and assessment of the results showed through digital maps within GIS-based environment. The main steps of the study are shown in Figure 1.

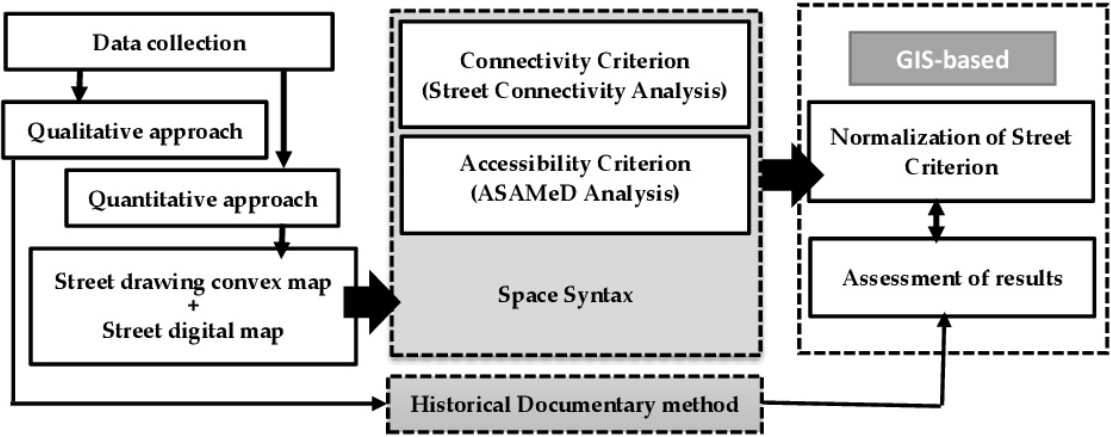

**Figure 1.** Methodological steps adopted in the study.

### 3.1. Data Collection and Analysis

Data was collected by using both qualitative and quantitative approaches. In fact, both approaches provide geographic data related to streets, public spaces, public buildings, etc. Moreover, all the streets were visited and analyzed, especially to collect some missing data.

In the first stage, by applying the historical documentary method, it was possible to reconstruct the urban evolution of the city and understand its current urban morphology [80,81]. For Qazvin study, by analyzing various materials and data, such as pictures, maps, and literature, the method identifies and classifies the traditional fabric's compounds

in terms of streets, squares and public buildings, which were used to build a map showing the urban morphology pattern and structure of the city.

In the last stage, streets were drawn in a digital map (GIS-based) as center lines, using the satellite view as the background. After that, the street network was simplified into a graph consisting of links in order to facilitate the space syntax geometric analysis [82]. Lines were converted into segments by clipping the initial lines at each intersection into smaller individual segments. Then, street segments were classified into two separate layers of the digital map (accessibility and connectivity). The two layers were based on the results obtained from the two analyses, i.e. ASAMeD and street connectivity.

### 3.2. Space Syntax Analysis

Space syntax was the process used to assess topological accessibility. Space syntax analyzes topological or relational aspects to understand how streets are connected and how they are accessed into a network. On a space syntax map, streets are represented by axial lines that correspond to the longest visibility line in a street, while segments are formed by chopping original axial lines into parts at each junction. In this study, space syntax was performed by using the DepthmapX software. This tool has been used by several authors in morphological urban studies [49,83]. DepthmapX allows an easy and fast data processing, does not requires special knowledge or technical expertise [41], and allows for a direct visualization of the spatial hierarchy of the street network. Therefore, DepthmapX can be used to visualize spatial relationships in a particular urban structure [83,84]. To achieve the study's goals, two different space syntax analyses were adopted: (i) ASAMeD; and (ii) street connectivity analysis.

ASAMeD was applied because often, the most accessible locations are not necessarily those closest to all other locations, but those closest in terms of topological turns and metrics [58]. In this sense, accessibility is related with the wayfinding skills and mental conceptualizations of the space people have [41]. ASAMeD analysis combined with two indicators including closeness and betweenness. It considers both indicators at the same time when applied. In fact, it shows that the analysis has two factors in mathematics formulation and then, both are coded in the one function.

Closeness is a syntactic measure of integration. Closeness measures how close each segment is to all other segments in the network. It describes how easy it is to get to one segment from all other segments. Closeness discriminates the layout design changes, measures the centrality, and detects spatial accessibility in terms of geometric simplicity/complexity. The question is how to differentiate a configuration where a pedestrian has an easy-to-grasp understanding to it and navigates the street network automatically [69]. Therefore, closeness is a measure suitable for identifying densities of located activities as well as determining an appealing location. Thus, closeness reproduces the to-movement potential of a spatial element as a destination [73]. Equation (1) describes the process of calculating closeness.

$$C_B\,(P_i) = \sum_j \sum_k P_{jk}\, g_{jk}\,(p_i) = g_{jk}(j < k) \tag{1}$$

In Equation (1), $g_{jk}(p_i)$ is the number of geodesics between node $p_j$ and $p_k$, which contain node $pi$ and $g_{jk}$, the number of all geodesics between $p_j$ and $p_k$.

Betweenness is a syntactic measure of choice that measures the through movement potential of a street segment following the least angular cost. It measures how many of the shortest paths between every pair of segments passes through each segment for all segments in the network [3]. Whenever a node is passed, the betweenness value is incremented. Equation (2) describes the process of calculating betweenness.

$$C_c\,(P_i) = \left(\sum_k d_{ik}\right)^{-1} \tag{2}$$

In Equation (2), $d_{ik}$ refers to the length of a geodesic (shortest path) between node $P_i$ and $P_k$. This equation reflects how close each segment is to all others under different types of distances.

In ASAMeD, the angular distance was calculated by applying a buffer with a spatial network distance of 300 m. In this buffer and for a given link, the overall links within a 300 m catchment were analyzed. The path with the least change in angle by way of the network to the initial link was taken while recording and summing the angular change along the path. This process was repeated for every link in the catchment. The radius of 300 m was used because it was defined as the most appropriate distance for access by walking to recreational facilities, public services such as schools and welfare centers [85], and transit stations [28]. Moreover, [15] found that most walking trips were less than 300 m.

In turn, street connectivity was applied as it analyzes how the street position is networked in a city [7,86]. Street connectivity can be understood as the directness and availability of alternative routes and can be defined as the number of intersecting streets per land-area unit [14,86]. High street connectivity is geographically found in areas with denser street networks and intersection density. A higher number of road intersections provide more potential routes for walking [86]. For these reasons, high street connectivity has an important impact on walking [57]. The process for calculating street connectivity is described by Equation (3).

$$Ci = \sum Kij \tag{3}$$

According to Equation (3), the centrality (*C*) of a street segment in a graph is defined by *Ci* (where *Kij* presents the direct bridge between units *i* and *j* is the number of direct connections to segment *i*).

As already mentioned, streets were drawn in digital maps and all the background. All the pedestrian and motorable streets were drawn as center lines. After that, the street network was simplified into a graph consisting of nodes and links in order to facilitate the geometric analysis [82]. Lines were converted into segments by clipping the initial lines at each intersection into smaller individual segments.

*3.3. Aggregation Method*

As the street connectivity and ASAMeD results were expressed in different quantitative scales, the values were normalized. The normalization process was implemented through fuzzy logic with a linear function, a common approach used, and decision problems [87]. The fuzzy theory is based on a fuzzy membership grade (possibility) that ranges from 0.0 to 1.0, indicating a continuous increase from non-membership to complete membership [78]. The calculation was estimated by using the sigmoidal function presented in Equation (4):

$$\begin{cases} x = x_{min} = 0 \ \rightarrow \ f(x_i) = 0 \\ x_{min} < x_i < x_{max} \ \rightarrow \ f(x_i) = \frac{x - x_{min}}{x_{max} - x_{min}} \\ x = x_{max} \ \rightarrow \ f(x_i) = 1 \end{cases} \tag{4}$$

where $X_i$ is the element of the network (*i* = 1, 2, . . . ., n) and *X* is the $X_i$ of elements of the network.

After normalizing the results, the next step is the aggregation of the relative values of ASAMeD and street connectivity. The highest aggregated values represent street segments with good performance on both approaches. In turn, the lowest aggregated values reflect bad performances in one or both approaches. Therefore, the final result of the assessment results from the aggregated values of street connectivity and ASAMeD, and assess the global conditions provided by the streets.

**4. Case Study: Qazvin (Iran)**

The study focused on Qazvin, which is a city in Iran. Qazvin provides a variety of streets (cul-de-sacs, gridded streets, local streets, collectors) and patterns from organic urban forms that have evolved since the middle ages to orthogonal grids in the newer neighbourhoods), which makes it interesting to assess the street network and its potential impacts on walkability. Qazvin is located in the province of Qazvin, 150 km northwest of

Tehran (Figure 2) and has 402,748 inhabitants. Qazvin is a medium sized city in Iran (Statistical Center of Iran, 2016). The city has experienced the fastest urban and demographic growth in recent decades due to its proximity to Tehran and the strategic location of the city that links the capital of the country to the northern, western, and Caucasian regions. In fact, Qazvin is one of the Middle Eastern cities belonging to the Silk Road. The city has more than 2000 architectural and archaeological sites, which reveals its antiquity. Archaeological excavations have confirmed the presence of sedentary communities in the Qazvin area since the Neolithic.

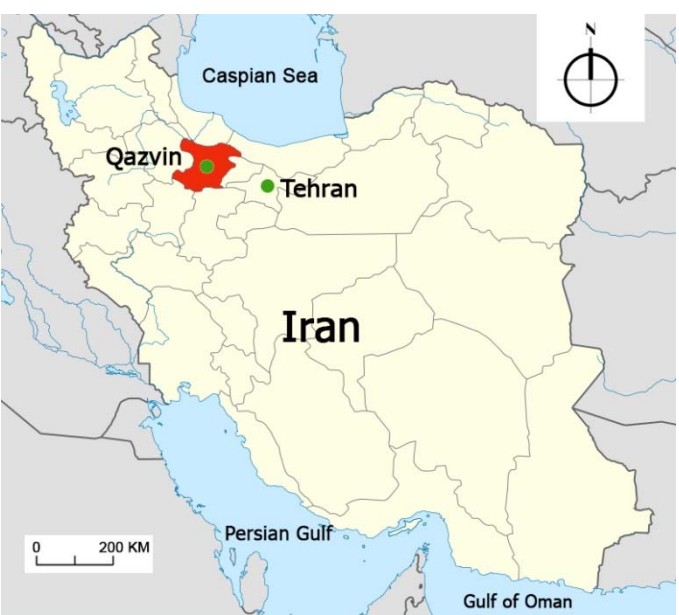

**Figure 2.** Location of the province of Qazvin in Iran.

### 4.1. Urban Structure of Qazvin

The focus of this subsection is to understand the urban structure of Qazvin through a documentary method and to show the normative framework of urban morphology pattern of the city, mainly the traditional fabric. Qazvin was founded and walled in the 4th century by King Shapur II of the Sassanid Dynasty. The city grew gradually inside the walls, but in the 7th century, it was captured by the Arabs. Qazvin flourished in early Muslim times, serving as a base for Islamization. Later, in the 13th century, the city was again conquered and sacked by Georgian, Mongolian, and Ottoman invasions. In the 16th century, Qazvin became the capital of the Safavid empire for half a century. This period left remarkable historical urban public spaces in the city including the governmental square and palace of Safavid, the governmental garden complex and official buildings, "Sepah" Street (one of the first streets in Iran), a bazaar, a great mosque, different religious centers, and "Sabzemeydan" (an urban gathering space). The connection of the new bazaar was around Chovgan Square and the old bazaar was alongside the square. Such structure strongly strengthened the socioeconomic development of the city.

Qazvin presents an urban structure that reflects an old and relatively unplanned process of urban growth. The city center dominates the morphological pattern of Qazvin where the mosque and several public facilities are located. Moreover, the city center is shaped by other minor centralities corresponding to neighborhood centers [88,89]. It is recognized that neighborhoods are influenced by the interaction between place and people and that they are well connected with the remaining urban areas [47,53,90]. Based on aerial photography of 1919 AD from Qazvin and the interpreted historical information [91], Figure 3 shows the neighborhood boundaries, center, and gates before the recent urban growth and transformation. At that time, the main urban structure of Qazvin was shaped by several sub-centers of neighborhoods, which are widespread in the city as a polycentric structure.

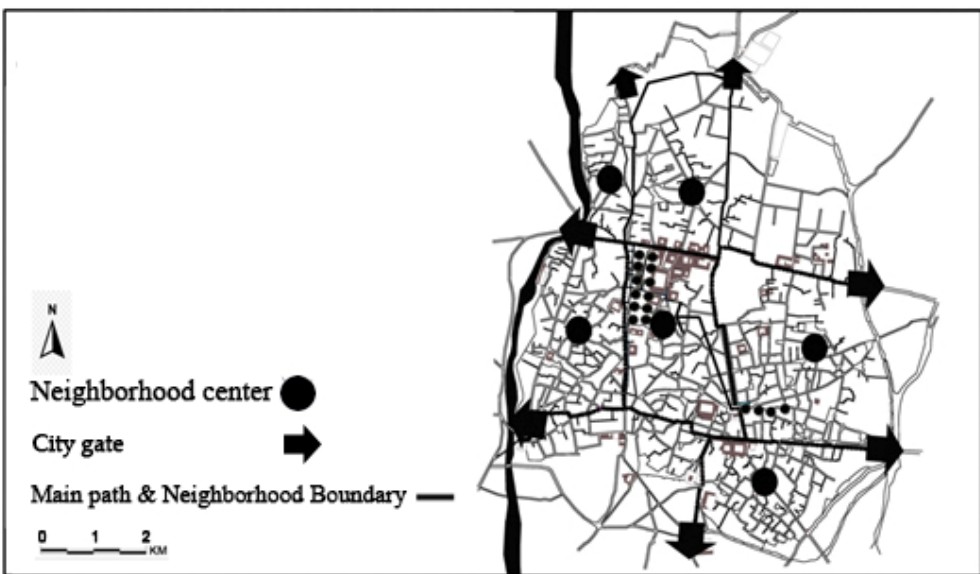

**Figure 3.** Analysis of the main Qazvin structure [91].

Polycentric structures are where activities are decentralized and are concentrated in neighborhood centers at the same time [45,92–94]. The neighborhood center is due to its hierarchical pattern; it has an important role in establishing a clear boundary between the public and private territories in urban space. Some researchers believe that such urban structures are distributed across the city as the traditional neighborhood has the potential to establish functional and sustainable spaces [95,96].

*4.2. Study Area delimitation*

The area of Qazvin selected for applying the methodology corresponds mostly to a part of the city center. The area encompasses 250 ha and corresponds to 15% of the urban boundary (Figure 4). Figure 4 shows the selected area (the blue polygon) to the full extent with black points in Figure 2 (1919 map), described in the legend. The selected area did not undergo major urban changes and comprises spaces with different urban origin functions and characteristics, linked by streets with different features.

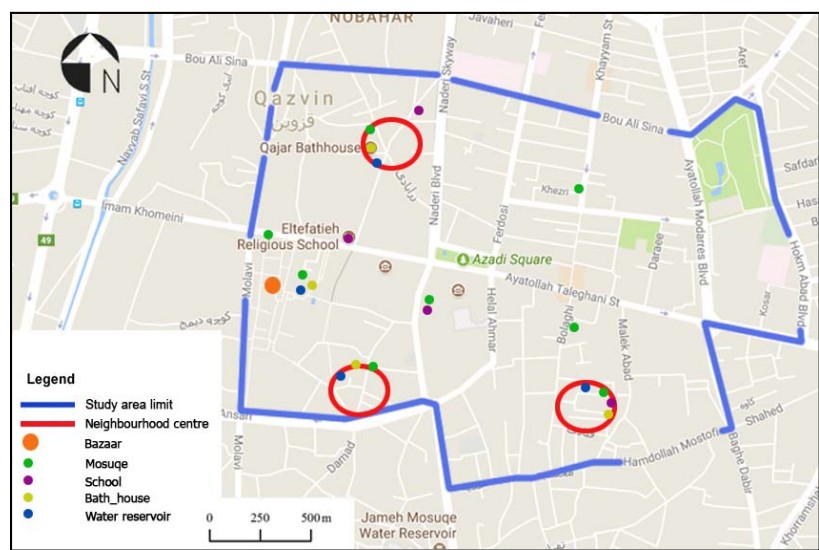

**Figure 4.** The study area in the city center of Qazvin.

The area shows the necessary conditions to implement the methodology in order to analyze the impact of the criteria in an urban context for walkability. The area integrates many

urban functions and organic urban morphology including history, aesthetics, and landscape values, which have different levels of accessibility. These different features provide a considerable variety of conditions for pedestrians, creating a varied and multifunctional space in the city.

Figure 5 shows the distribution of the streets according to their length. As can be concluded, in the center of Qazvin, most street lengths are shorter than 200 m including 94% of streets, which means that there is an organic form with high compression. These streets belong to traditional neighborhoods, and almost all without sidewalk space. Additionally, the average street's width is less than 5 m and the slope average is around zero.

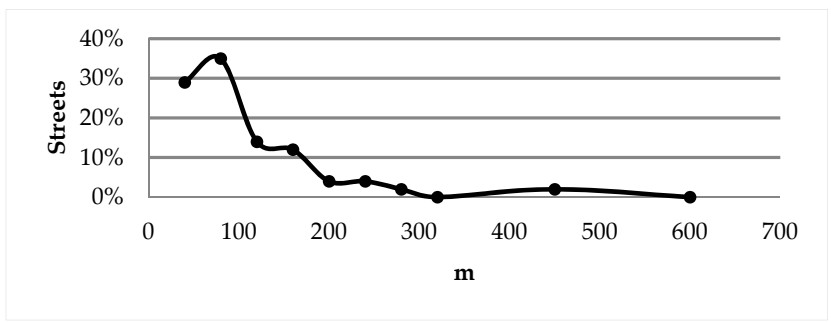

**Figure 5.** Street length in the selected area of Qazvin.

## 5. Results

This section presents the results obtained in the study by applying the above-mentioned methodology to the selected area of Qazvin. The main features of the space syntax analysis are presented in Table 1. In terms of connectivity, 566 axial lines compose the streets on the chosen area. The number of street connections changes between 1 and 21 (Table 1 and Figure 6A), meaning there are streets with only one connection until the street with 21 connections. As can be concluded in Figure 6A, most streets located in this central area of Qazvin only have less than three connections with other streets. This is directly related to the urban structure of the city core, where many streets are of local access. In this structure, residential units form their own neighborhoods as they grow in size and create their own social unit. In most cases, each neighborhood was defined at the perimeter through the hierarchy pattern of streets. As can also be concluded from Figure 6A, the blue axial lines, corresponding to streets with less connections, were the most dominant in the street connectivity map. According to Table 2, these streets are 58.2% of the total length. The streets with more connections are represented by red and they mostly correspond to longer streets such as Ayatollah Taleghani, Ferdosi, Imam Khomeini Streets, and Naderi Boulevard. As denser connectivity provides more potential routes for walking and has a positive impact on walking, the streets with more connections were ranked with higher scores than those less connected. Accordingly, as shown in Table 1, the normalized values assigned to the number of connections ranged from 0 (for dead-end streets with only one connection) to 1 (streets with 21 connections, the maximum identified in this central area of Qazvin). The prevalence of dead-end streets is strictly related to the presence of neighborhoods and can be confirmed by the street-length pattern found in the central area of Qazvin.

The main features of ASAMeD are presented in Table 1 and in Figure 6B. This space syntax analysis divided the streets within the delimited area into 2053 segments. There were more segments than axial lines because the tool cuts the original axial lines into parts at each junction. The axial lines were divided in a number of segments ranging from six to 127. This ranking arises from the sum of angular changes along the path. Every segment was ranked regarding the number of k segments in each catchment. Thus, the streets with higher scores were the most accessible or those easier to get to from all other places. As shown in Figure 6B, the streets highly scored defined widespread sub-centers in the selected area of Qazvin. These areas define attractive and accessible urban centralities

widespread by the city centers that correspond to neighbourhoods. These were the cases of the Bazaar and the neighbourhoods of Darb-Kooshek, Bolaghi, and Sar-kocheh Reyhan. Thus, this measure can reproduce the distribution of attractiveness and the actual density of the located activities. Regarding Table 3, in most of the analyzed streets, segments had an intermediate level of accessibility (green) encompassing the neighborhood centers with 63.8% from the total length. The most peripheral segments were globally less scored (blue segments).

**Table 1.** Normalization of street connectivity and ASAMeD.

| Street Connectivity | | | | ASAMeD | |
|---|---|---|---|---|---|
| Score | Normalized Values | Score | Normalized Values | Score | Normalized Values |
| 1 | 0 | 12 | 0.55 | 1–25 | 0.05 |
| 2 | 0.05 | 13 | 0.6 | 26–50 | 0.25 |
| 3 | 0.1 | 14 | 0.65 | 51–75 | 0.45 |
| 4 | 0.15 | 15 | 0.7 | 76–100 | 0.65 |
| 5 | 0.2 | 16 | 0.75 | 101–125 | 0.85 |
| 6 | 0.25 | 17 | 0.8 | 126–150 | 1.00 |
| 7 | 0.3 | 18 | 0.85 | | |
| 8 | 0.35 | 19 | 0.9 | | |
| 9 | 0.4 | 20 | 0.95 | | |
| 10 | 0.45 | 21 | 1.00 | | |
| 11 | 0.5 | | | | |

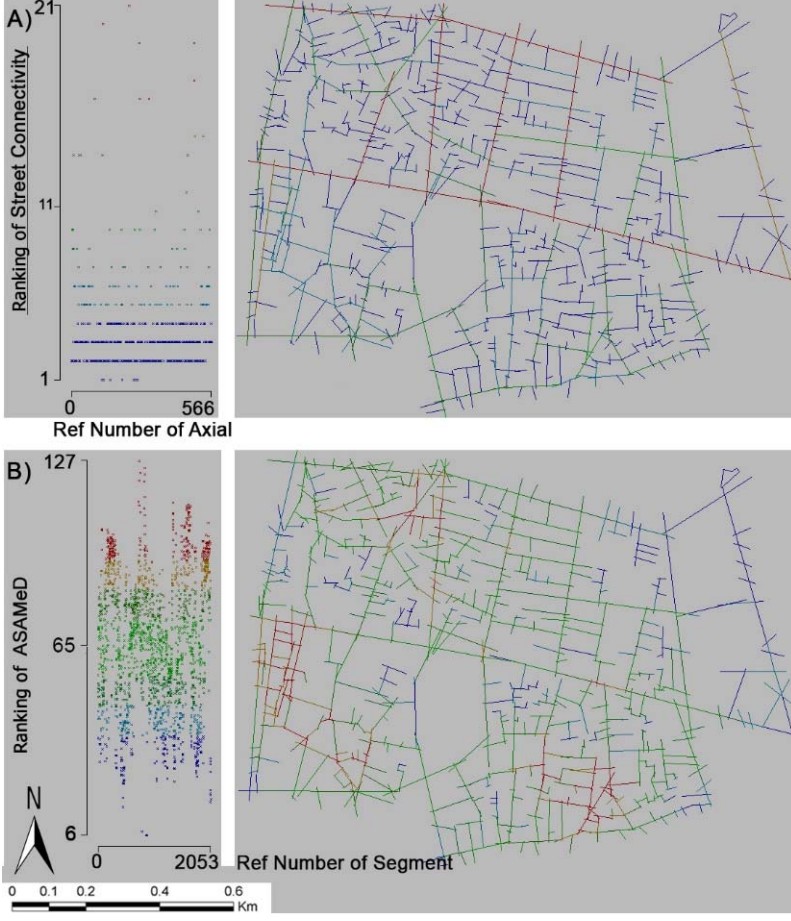

**Figure 6.** (**A**) Street connectivity and (**B**) angular segment analysis by metric distance (ASAMeD).

Figure 7 shows the streets of Qazvin according to the levels of connectivity. The highest connectivity is found in the streets of Nadri, Ferdowsi, Bazar, and Bu-Ali-Sina, which mainly cross and surround ancient neighborhoods. Also, the distributed streets

length that got the higher rank average is 25.5 percent in the Table 2. Another comparison between streets with yellow and red colors, streets with lower levels of connectivity were in the south area of the city. This part has been affected by new urbanism and the implementation of new urban planning during recent years, where the municipality of Qazvin damaged the traditional neighborhoods by cutting the urban structure with a direct street called "Ansari Street".

**Table 2.** Disaggregated result of the Street Connectivity ranking.

| Street Connectivity Classes | Total Length (m) | Percentage |
|---|---|---|
| 0.0–0.249 | 26,600.9 | 58.5 |
| 0.25–0.499 | 7311.7 | 16 |
| 0.50–0.749 | 5915.1 | 13 |
| 0.75–1.00 | 5693.7 | 12.5 |

**Table 3.** Disaggregated result of the ASAMeD street ranking.

| ASAMeD Classes | Total Length (m) | Percentage |
|---|---|---|
| 0.0–0.249 | 5336.6 | 11.7 |
| 0.25–0.499 | 29,041.7 | 63.8 |
| 0.50–0.749 | 8210.8 | 18 |
| 0.75–1.00 | 2932.3 | 6.5 |

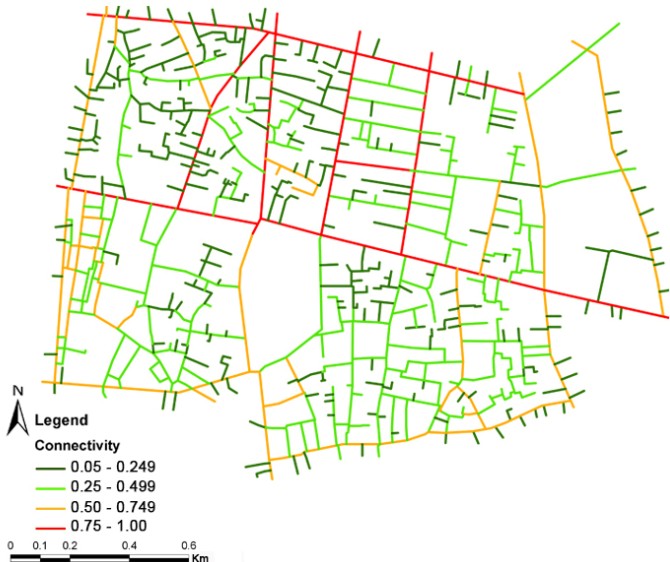

**Figure 7.** Street ranking based on street connectivity by GIS (Qazvin).

Figure 8 shows the streets of Qazvin according to the levels of topological accessibility (ASAMeD). According to this analysis, the highest values of integration and choice are found in old neighborhood centers (Darb Kooshek neighborhood, Sar-kocheh reyhan neighborhood, Bolaghi neighborhood). Thus, these older centers are the most accessible and topologically closer in relation to the remaining streets. Also, the distributed streets length that got the higher rank average is 24.5 percent shown Table 3. Moreover, this analysis explains that the structure created the special pedestrian movement pattern.

Figure 9 shows the final result combines the street connectivity and ASAMeD results and this result considers that only the highest ranking streets are obtained by summing the results of each analysis. The highest ranking was more than 0.5 from the two analyses and is presented by a score of 1 in the legend of the new map. This result shows the pedestrian network that it has influenced from the urban structure in terms of the amount of closeness and betweenness to the pedestrian network. The highest rankings of streets from both

analyses were 66.9 percent from the total length distribution in Qazvin shown in Table 4. These streets made the cohesive pedestrian network and covered the whole area, and even minor-centers of the neighbourhood. Furthermore, Figure 9 shows that the centers have joined to the pedestrian network, represented by blue circles. The area includes three neighbourhoods named Drab-Kooshek, Bolaghi, amd Sar-kocheh Reyhan and the tradition market (Bazaar). Additionally, each neighbourhood provides traditional infrastructures that contain schools, mosques, baths, and Ab-anbar in the center [91].

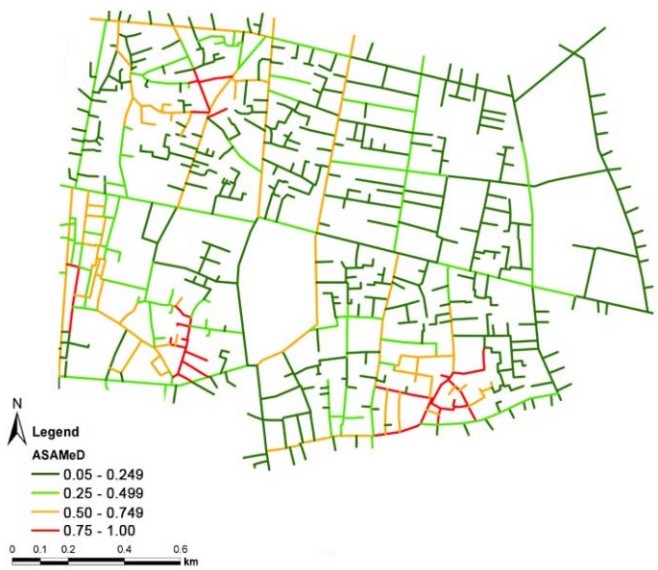

**Figure 8.** Street ranking based on ASAMeD by GIS (Qazvin).

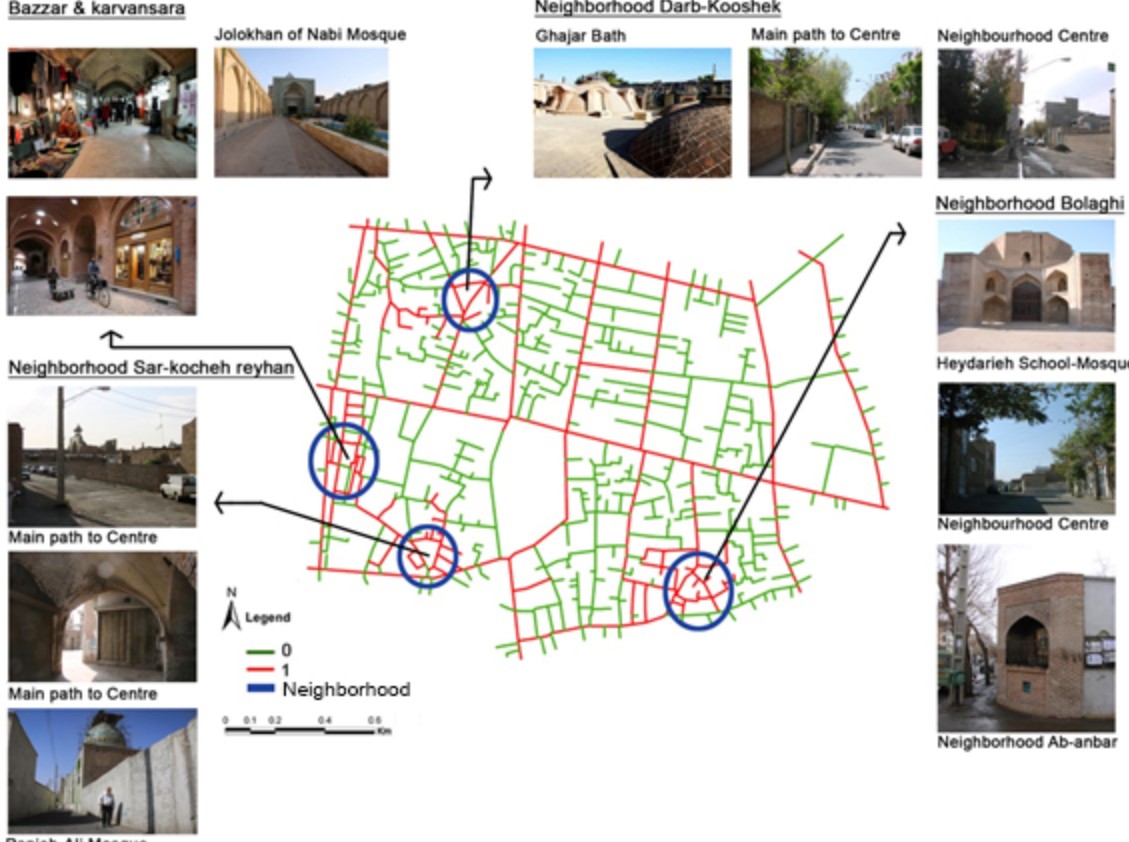

**Figure 9.** Highest ranking from combining the ASAMeD and street connectivity results (Qazvin).

**Table 4.** Disaggregated result of the final street ranking.

| Pedestrian Network Classes | Total Length (m) | Percentage |
| --- | --- | --- |
| 0 | 15,030.7 | 33.1 |
| 1 | 30,490.7 | 66.9 |

## 6. Discussion

This research aimed to understand the impact of the urban morphology on walkability and can be used to evaluate and improve pedestrian networks. The method assessed the street network with two main criterion (connectivity and accessibility) by applying the space syntax and the documentary approaches. Space syntax theory reflects pedestrian decisions as perform spatial analysis of interactions and in this research explains the relationships between topology and geography in walkability. In this context, by using space syntax, the study measured two criteria: angular segment analysis by metric distance—ASAMeD and street connectivity. The ASAMeD identifies potential streets that pedestrian use for walking and the street connectivity shows the spatial configuration of the network.

The impact of urban morphology on walking has been well-documented by several researchers in this field [34,54,97,98]. More specifically, the spatial distribution of buildings and the arrangement of streets are seen as having a strong influence on the means and intensity of human activities, social interactions, spatial mechanisms, pedestrian behaviors, and human activities. The findings presented in this research show that a well-functioning urban structure should have accessible and connected neighborhoods, where activity centers are within a convenient walking distance. The findings also showed that cities with polycentric urban morphologies could be more attractive for pedestrians, if these local centers are well accessible and interconnected between them, as found in the city center of Qazvin. In the past, Qazvin's traditional urban morphology was supported by a walking structure around the old town in the central area of the city and through hierarchical streets. Most of these streets were pedestrian-friendly and urban spaces suitable for pedestrians with suitable shade during the summer. But nowadays, most peripheral streets are globally less adapted to walking.

The practical implication of the study findings are in identifying what the type of urban structure is and how to affect the linkage with the pedestrian network. The ASAMeD results showed that the streets network in Qazvin could attract pedestrian movement in a centralized way focus in the center of each neighborhood. In fact, the neighborhood center areas, as one of the main structural elements of the streets network, has created the widespread structure in the city. In contrast, the ASAMeD results did not have any linkages with streets that received higher scores in the connectivity analysis. The result of the street connectivity also showed in the south area of the city, which exhibited scores significantly lower than 0.75 (the maximum score achieved was between 16 and 21 in Figure 5, and the final street ranking after normalization was between 0.75 and 1 in Figure 7). In the model's last stage, which entailed combining the highest-ranking of streets obtained from both analyses, the output showed that there was a cohesive network (red lines in Figure 9) in the organic urban morphology. This means that it is a pedestrian network developed by the polycentric structure and will be improved by the cohesion into the pedestrian network. As illustrated in Figure 9, the two main criteria were addressed to assess a cohesive network. In addition, it will be more effective to create a livable neighborhood based on a pedestrian movement pattern and spatial relationship.

The global scores of accessibility and connectivity found in the central area of Qazvin are acceptable. The analysis suggested that highly-integrated and connected streets may facilitate walking, partly through the availability of attractive destinations widespread by the city center, namely in the neighborhoods, confirming that well-connected and accessible streets are more conducive for walking. City stakeholders have to be considered with a cohesive pedestrian network as a guiding intervention policy in urban space that will

improve the quality of life in the city for citizens. Moreover, it will figure out the main lines for a comprehensive plan for sustainable development mobility.

## 7. Conclusions

This research aimed to study the relationships between urban morphology and walkability by combining two space syntax and topological measures. The method assessed the condition of connectivity and accessibility provided by streets network to pedestrians' walkability by applying space syntax analyses and the historical documentary technique. The method was applied in the city center of Qazvin, Iran. The study considered in the assessment process, at the same time, how the streets and how the network attract pedestrian movement. In the plan of the city, streets with more potential to attract pedestrian contributes to creating places that are more desirable. Additionally, this study noticed relationships between spatial layout and a range of social and environmental phenomena through a configurational analysis of the urban street network of Qazvin. In fact, the methodology presented in the study communicated between multi-scales that the relationship is between the street network and urban centers and neighborhoods. These findings can help researchers, city planners and designers to develop an in-depth and understanding of how the urban morphology may influence walkability.

The study demonstrates that geographical and topological measures describe the street network in different ways. The findings are clearly showed in the simultaneous analyses of Figures 7 and 8: the streets more interconnected are not the more integrated and vice-versa. The combination of both approaches proposed in this study gives a different overview about how pedestrian-friendly the streets may be and provides new insights into how to promote cohesive pedestrian networks. Moreover, the combination of street connectivity and ASAMeD captured not only spatial aspects of the street network of Qazvin, but also functional aspects, such as the presence of specific facilities and main destinations (Bazaar, mosques, schools, etc.).Thus, designing a pedestrian network based on polycentric urban structures increases the walkability in the entire city and, in addition, helps to control urban sprawl as reported in various studies [45,99,100]. Hence, between the main types of urban structure, the polycentric structure makes a strong pedestrian network that can increase cohesion of the pedestrian network and prevent city sprawl, and support planning policies to improve the conditions provided to pedestrians.

The described method has its own limitations. Firstly, the evaluation was mostly supported on geometric and topological characteristics of the street network. Thus, the evaluation shows the most connected and integrated streets and urban spaces but does not reflect other important criteria having impacts on walking, such as the characteristics and conditions of sidewalks, traffic safety, land use mix and security. For example, it is recognized that a well-connected street may not be attractive for pedestrians if sidewalks are narrowed, poorly maintained, and hillier or without safer crossings. This means that the evaluation method described in this study should be complemented and articulated with other approaches to include other built environment and streetscape variables with impacts on walking. Secondly, the findings described in this paper are not representative as they reflect the specific characteristics of the urban morphology of Qazvin. In the future, more studies should be conducted to evaluate the impact, especially of other urban morphologies (such as orthogonal urban structures), on the attractiveness of pedestrian streets.

**Author Contributions:** Conceptualization, M.J.; methodology, M.J.; software, M.J.; validation, M.J., and R.R.; formal analysis, M.J.; investigation, M.J. and F.F.; resources, M.J.; data curation, M.J.; writing—original draft preparation, M.J.; writing—review and editing, M.J. and F.F.; visualization, M.J. and F.F.; supervision, R.R.; project administration, R.R.; funding acquisition, R.R. All authors have read and agreed to the published version of the manuscript.

**Funding:** This research was funded by the JPI Urban Europe and FCT, grant number ENSUF/0004/2016.

**Institutional Review Board Statement:** Not applicable.

**Informed Consent Statement:** Not applicable.

**Acknowledgments:** The study partially summarizes the results of a PhD thesis entitled "Combining Multi-Criteria and Space Syntax Analysis to Assess a Pedestrian Network: An Application for the Pedestrian Network of Porto (Portugal) and Qazvin (Iran)", supervised by Rui Ramos, the Center for Territory, Environment, and Construction of the School of Engineering of University of Minho. ''SPN: Smart Pedestrian Net" Project is special funding provided by JPI Urban Europe titled Smart Urban Futures; and co-financing by FCT-Fundação para a Ciência e Tecnologia (ENSUF/0004/2016).

**Conflicts of Interest:** The authors declare no conflict of interest.

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
