# Peer review of "Accessibility and Connectivity Criteria for Assessing Walkability: An Application in Qazvin, Iran"

_sustainability, doi:10.3390/su13073648_

Round 1
Reviewer 1 Report
This is a case study where the relationships between urban morphology and the choice of path by pedestrians are explored for a city in Iran. The reviewer has the following concerns.
1. The title of the article is very misleading. In the current format, it reads more like a literature review. However, this article is just a case study.
2. The major concern is regarding the contribution to the current body of knowledge. The problem explored is not now. Furthermore, all the matrices used to evaluate road segments are borrowed from other studies. Lastly, the findings do not present any new information. The findings reported using the case of a single city would not be proper to draw meaningful conclusions.
Reviewer 2 Report
The paper is very well structured. In detail describe carried scientific research and its results. It has aim to explain relationships between topology and geography to the street network. It is based to the known methodology, which is tested through case city.
However, the paper should reconsider some changes and improvements:
- The references are not numbered in square bracket in text.
- There is a very large number of “References” which should be shortened.
- In the paper is often used “study” and should be “paper”
- All equations are numbered in brackets. But, those numbers should be placed on the right margin of the text
- Text font in tables and figures should be according to the instructions for the authors
- In the subchapter 4.1 should be given wider explanation of urban structure of the case city (not only description of the city from 1919)
- In the subchapter 4.2 should be wider description of study area limit and the reasons of that choice
Reviewer 3 Report
It appears that this paper focuses on the link between built environment on route choice made by pedestrians. This study makes use of space syntax with a focus on two criteria : connectivity and accessibility. The paper also appears to be well structured (introduction, discussion, case study, method, results and conclusion). Furthermore, the illustrations are of good quality and well integrated/commented in the text.
However, I am sorry to say that, in its present state, I cannot evaluate this paper because of a substandard quality of writing/use of English. Sentences/words are misused, as well as tenses, while other occurrences appear to have been directly translated with an automatic translator. As a whole, it makes the text quite incomprehensible. The writer is not a native speaker, which is obviously not an issue by itself. However, when a writer is not a native speaker, the whole point is to reach a certain threshold in term of quality of writing in which the reader will not wonder if the writer is a native speaker. This threshold is not reached here, since it is quite clear from the very first sentence of the abstract that the writer is not at ease with English.
I do not question the scientific quality of this paper. However, I ask the authors to first improve the overall quality of writing by using, for example, an official proofreading/translating service.
Reviewer 4 Report
This work presents a topic of interest and current relevance in the understanding of the factors that affect the vitality of the urban scenes, especially in old and complex urban environments like Qazvin.
The global approach of the manuscript is adequate and contains the necessary sections, with a logical and systematic development of the contents. The bibliographic review is quite consistent and complete, However, it is recommended to take into account some basic references on the subject in order to strengthen the theoretical and methodological point of view:
Ewing, R., Cervero, R., 2010. Travel and the built environment. Journal American Planning Association, 76:265–294. http://dx.doi.org/10.1080/01944361003766766
Marquet, O., Miralles-Guasch, C., 2014. Walking short distances. The socioeconomic drivers for the use of proximity in everydaymobility in Barcelona. Transportation Research Part A: Policy Practice, 70:210–222. http://dx.doi.org/10.1016/j.tra.2014.10.007
Ståhle, A., Marcus, L., Karlström, A., 2005. Place Syntax - Geographic Accessibility with Axial Lines in GIS. Proc. 5th Int. Sp. Syntax Symp, 131–144. http://spacesyntax.tudelft.nl/media/Long%20papers%20I/stahle.pdf
Vale, D.S., Saraiva, M., Pereira, M., 2015. Active accessibility: a review of operational measures of walking and cycling accessibility. Journal of Transport and Land Use, 1:1–27. http://dx.doi.org/10.5198/jtlu.2015.593
van Wee, B., 2016. Accessible accessibility research challenges. Journal of Transport Geography, 51:9–16. http://dx.doi.org/10.1016/j.jtrangeo.2015.10.018
In addition, I would like to make a few suggestions to complete the job.
1. Introduction
It would be necessary to identify some references to a fifth area of ​​interest for research that has not been taken into account (lines 37-40): Mobility system, urban design and planning. Some suitable references could be:
Arranz-López, A., Soria-Lara, J.A., López-Escolano, C., Pueyo Campos, A. 2017. Retail Mobility Environments: A methodological framework for
integrating retail activity and non-motorised accessibility in
Zaragoza, Spain. Journal of Transport Geography, 58:92–103. http://dx.doi.org/10.1016/j.jtrangeo.2016.11.010
Cerin, E., Leslie, E., du Toit, L., Owen, N., Frank, L.D., 2007. Destinations that matter: associations with walking for transport. Health Place, 13:713–724. http://dx.doi.org/10.1016/j.healthplace.2006.11.002
Southworth, M., 2005. Designing the Walkable City. Journal of Urban Planning and Development, 131:246–257. http://dx.doi.org/10.1061/(ASCE)0733-9488(2005)131:4(246)
Talavera-Garcia, R., Soria-Lara, J.a., 2015. Q-PLOS, developing an alternative walking index. A method based on urban design quality. Cities, 45:7–17. http://dx.doi.org/10.1016/j.cities.2015.03.003
4.2 Study area limit
In this section, it is necessary to address other relevant issues for pedestrian mobility in relation to the characteristics of the streets: average width, average length, slope, state of conservation, existence of sidewalks, existence of obstacles that hinder pedestrian flows, etc. This information is relevant to later understand and contextualize the results obtained.
Figure 4: Font of the basemap?
5. Results
Figure 9: What is 0 and what is 1 in the Map Legend?
6. Discussion and final remarks
It is necessary to make a better interpretation of the findings obtained after applying the analysis method with respect to the particularities of the analyzed area of ​​the Qazvin city center. This should serve to guide intervention policies in urban space aimed at improving and strengthening sustainable mobility options (especially pedestrian) in this area.
Author Response
Dear Reviewer
Please check the file in the attachment.
Best regard
Mona Jabbari
